# Interdependencies of the Neuronal, Immune and Tumor Microenvironment in Gliomas

**DOI:** 10.3390/cancers15102856

**Published:** 2023-05-21

**Authors:** Alexander Yuile, Joe Q. Wei, Aditya A. Mohan, Kelly M. Hotchkiss, Mustafa Khasraw

**Affiliations:** 1Department of Medical Oncology, Royal North Shore Hospital, Reserve Road, St Leonards, NSW 2065, Australia; 2The Brain Cancer Group, North Shore Private Hospital, 3 Westbourne Street, St Leonards, NSW 2065, Australia; 3Sydney Medical School, Faculty of Medicine and Health Sciences, The University of Sydney, Sydney, NSW 2006, Australia; 4The Preston Robert Tisch Brain Tumor Center, Duke University, Durham, NC 27710, USA

**Keywords:** glioma, glioblastoma, astrocytoma, tumor microenvironment, immune microenvironment

## Abstract

**Simple Summary:**

Gliomas are the most common primary brain tumors. These cancers are universally fatal with limited treatment options. Glioma cells co-opt non-cancerous cells present in normal brain tissue. This manipulation results in a complex network of cell interactions. This interplay is further complicated by variations depending on specific mutations in glioma cells. In order to identify future treatments for gliomas, a better understanding of these interactions is needed. To address this, we review the literature to highlight these interactions and how they relate to different glioma mutations.

**Abstract:**

Gliomas are the most common primary brain malignancy and are universally fatal. Despite significant breakthrough in understanding tumor biology, treatment breakthroughs have been limited. There is a growing appreciation that major limitations on effective treatment are related to the unique and highly complex glioma tumor microenvironment (TME). The TME consists of multiple different cell types, broadly categorized into tumoral, immune and non-tumoral, non-immune cells. Each group provides significant influence on the others, generating a pro-tumor dynamic with significant immunosuppression. In addition, glioma cells are highly heterogenous with various molecular distinctions on the cellular level. These variations, in turn, lead to their own unique influence on the TME. To develop future treatments, an understanding of this complex TME interplay is needed. To this end, we describe the TME in adult gliomas through interactions between its various components and through various glioma molecular phenotypes.

## 1. Introduction

Gliomas are the most common primary brain malignancy [1]. Since 2016, the WHO Tumor classification has dichotomized gliomas by the presence or absence of an *IDH1/2* mutation [2]. The wildtype IDH1 enzyme and IDH2 enzymes (encoded for by *IDH1* and *IDH2*, respectively) convert isocitrate to α-KG. In the presence of an *IDH1/2* mutation, these enzymes produce 2-hyroxyglutarate (D-2-HG) from isocitrate instead [3,4,5,6]. D-2-HG causes a cascade of pro-oncogenic events [6,7]. In the current WHO 2021 classification, *IDH1/2*-mutant gliomas are further classified into those with a 1p/19q codeletion (oligodendrogliomas grade 2–3) and those without (astrocytomas grade 2–4). Astrocytic gliomas with high-grade features (either histopathologic or molecular) without an *IDH1/2* mutation are termed *IDH1/2* wildtype glioblastomas [3].

This adoption of molecular classification signifies an understanding that gliomas with different molecular characteristics behave differently. Compared to wildtype glioblastomas, patients with IDH-mutant astrocytomas are younger at diagnosis and have longer survival. In addition, there is a growing appreciation that this divergent clinical behavior is linked to a complex interdependent relationship with the tumor microenvironment (TME) that varies based on molecular phenotype. Compounding this further is the growing appreciation of non-tumoral, non-immune cells in the TME dynamic, including neuronal and glial modulation of the TME [8].

A better understanding of TME interactions and the impact of molecular phenotype is a crucial step to developing further treatments. For example, gliomas are termed immunologically “cold” because of the predominant immunosuppressive interplay of the TME [9]. This has resulted in immunotherapy being ineffective in gliomas. Immunotherapy has been relatively effective in cancers such as melanomas [10,11], but immunotherapy trials in gliomas have yet to produce positive clinical trial results [12]. 

If the complex and dynamic web of tumor, immune and non-tumor, non-immune interactions can be fully elucidated, then treatments can be used to prime the TME to be more vulnerable to therapies such as immunotherapy. Here we describe the contents of the adult glioma TME in tumoral, immune and non-tumoral, non-immune components and explore the interactions between these groups. We describe how glioma phenotypes alter these interactions and explore novel therapeutic options arising from current TME understanding. 

## 2. Tumor Components of the TME

Glioma cells exist in a network with cells more central to the tumor mass being connected by a synaptic network, with invading tumor cells at the periphery being unconnected [13]. The tumor cell components of the TME comprise stem cells like glioma cells and differentiated glioma cells. The latter forms a heterogenous group defined by differing molecular and transcriptomic signatures [14]. 

### 2.1. Glioma Stem Cells

These precursor cells exist in the perivascular niche, forming a glioma stem cell pool [15,16]. The cancer stem cell model argues that a self-renewing population of progenitor cells allows for differentiation into heterogeneous cancer cell populations [17]. Glioma stem cells (GSCs) are widely considered to be resistant to therapy and allow for tumor cell repopulation after therapy [18]. In addition, they have been shown to be extremely plastic, providing a variety of glioma clonal populations and differentiating into supportive cells such as vascular endothelial cells [5,17]. These factors may explain why there is a positive correlation to stem cell population and tumor grade [4,19]. 

### 2.2. Mutational Landscape of Glioma Cells 

In general, the most common aberrant molecular pathways are the receptor tyrosine kinase (RTK) pathways (which are further divided into the MAPK-pathway and the AKT/mTOR-pathway), the RB-pathway, and the p53-pathway. Alterations in these pathways tend to be mutually exclusive, and glioma cells tend to harbor an alteration in each of these pathways [6,20,21]. This suggests a highly interactive network of molecular alterations.

Reflecting this interplay, it has been recognized that there are recurring patterns in the molecular and mutational landscape. For example, abnormalities of copy number variants occur at a much higher frequency than specific mutations, with deletions having a higher prevalence than amplifications [21]. The most common deletions involve the *CDKN2A* gene of the RB-pathway, *PTEN* of the AKT/mTOR-pathway, and *NF1* of the MAPK-pathway. While the most common amplifications include the RTK receptors *EGFR* and *PDGFRA*, *MDM2* and *MDM4* of the p53-pathway, *PIK3CA* of the AKT/mTOR-pathway, and *CDK4/6* of the RB pathway, the most frequently observed hotspot mutation is the *TP53* gene encoding for p53. Other commonly mutated genes include *PTEN*, *NF1*, and *EGFR (EGFR* mutations usually occur with exon 1–8 aberrancy, which is referred to as *EGFRvIII*) [20]. 

Further complicating this landscape, the glioma mutational phenotype is heterogenous and highly plastic. For example, patterns of gene mutations tend to only be seen on recurrence, such as *NF1* and *TP53* co-mutations, *Rb1* and *PTEN* co-deletions, and *LTBP4* gene abnormalities [22]. This clonal change bares consideration as it, in turn, can modify the tumor microenvironment. For example, *LTBP4* mutations have been shown to upregulate TGF-beta, which has significant anti-inflammatory properties (discussed below). It is also now recognized that mutational switching occurs in key mutational pathways, such as RTK and p53 pathways, where one pathway mutation is replaced by another on recurrence [6,22,23].

### 2.3. Tumor Cell Subtypes

In addition to molecular features such as IDH status, glioma cells can be classified by their transcriptional profile. Consensus clustering has been used to describe four classes using 840 classifying genes (210 genes per class) [6]. These subtypes, based on prior descriptions and their expression signatures, were named proneural, neural, classical, and mesenchymal [6] (Figure 1). 

#### 2.3.1. Proneural

Proneural subtype gliomas are so named as they have overexpression in multiple proneural development genes such as the *SOX* genes. They also involve genes associated with development and cell cycle/proliferation [24,25]. 

Glioma cells meeting the proneural signature most commonly had *IDH1/2* mutations, as well as focal amplification of RTK receptor gene *PDGFRA* [6,22]. This *PDGFRA* amplification is seen almost exclusively in this subtype. Where *PDGFRA* genes are unaltered, proneural gliomas almost always have increased activity of the genes *PIK3CA* or *PIK3R1* [6]. Unlike classical subtype gliomas, proneural gliomas are also commonly associated with *TP53* mutations [6].

Proneural glioma cells have an increased expression of oligodendrocyte development genes, such as the aforementioned *PDGFRA*, as well as other markers, such as *NKX2* and *OLIG2* [26]. Adding evidence to the oligodendrocyte phenotype is the expression of SOX10, encoded by a SOX gene, which can induce oligodendrocyte differentiation by antagonizing the NFIA protein. Conversely, NFIA can also inhibit SOX10, leading to astrocyte differentiation. Given SOX genes are overexpressed in the proneural subtype, it is understandable this subtype has an oligodendrocyte-like phenotype [27]. 

Interestingly, *OLIG2* suppresses p21, which is an apoptotic regulator of the p53 pathway [28]. The impact of *OLIG2* is an example of how the expression of these oligodendrocyte-associated genes are themselves tumorigenic. 

Given the strong association between *IDH1/2* mutations and the proneural subtype, molecular differences between the proneural subtype compared to other subtypes also serve as a description of the molecular phenotype associated with *IDH1/2* mutations.

#### 2.3.2. Classical

Glioma cells meeting the “classical” expression pattern were noted to have chromosome 7 amplification and chromosome 10 loss at a high frequency (almost 100% in classical glioblastomas). In addition, most cells had *EGFR* amplification and homozygous deletion of *CDKN2A* but were lacking *TP53* and *IDH* mutations and *PDGFRA* and *NF1* alterations when compared to other subtypes [6]. Other mutations commonly seen in classical type gliomas include those of the NOTCH pathway, such as *NOTCH3*, *JAG1*, and *LFNG*, and in the Sonic Hedgehog pathway, such as *SMO*, *GAS1*, and *GLI2*. This subtype is also associated with increased expression of neuronal stem cell markers [6]. 

#### 2.3.3. Mesenchymal

This subtype most notably has *NF1* alterations (predominantly hemizygous deletions at 17q11.2), resulting in lower *NF1* expression. There is also increased expression of genes of the tumor necrosis superfamily and NF-κB pathways, such as *TRADD, RELB*, and *TNFRSF1A* [6,23]. The increased expression of these genes may explain, in part, why this subtype is the most inflammatory with the highest immune cell burden (discussed below). 

#### 2.3.4. Neural

The “neural” subtype most commonly has expression of typical neuronal markers such as *NEFL*, *GABRA1*, *SYT1*, and *SLC125A5*. Genes associated with this subtype are commonly related to neurons, such as those involved in neuronal axial development, neuronal projection, and synaptic transmission [6].

## 3. Immune Components of the TME

A significant proportion of the glioma tumor mass is comprised of immune cells. Our immune system can be generally divided into innate and adaptive arms (Figure 1). The innate immune system provides a rapid response to pathogens but lacks antigenic specificity and immunological memory [7]. The adaptive immune response can mount antigen-specific responses and form memory immune cells [29]. Upon antigenic restimulation of the same antigen, the immune response can increase in speed and magnitude. Both arms are thought to be important in immune surveillance and preventing immune escape by cancerous cells [30]. Although it is long believed that innate and adaptive arms of the immune system are separate entities, there is mounting evidence that there are interplays between the two, generating complex immune responses and forming long-lasting memory cells. Furthermore, immunosuppressive cells play a crucial role in dampening down immune responses to prevent overreaction, especially in the setting of autoimmunity.

Immune cells found in the glioma TME include macrophages, neutrophils, dendritic cells (DCs), and natural killer (NK) cells of the innate immune system; CD4+ T cells, CD8+ T cells, and B cells of the adaptive immune system; and immunosuppressive cells such as monocyte-derived suppressor cells (MDSCs) and regulatory T cells (Treg). 

### 3.1. Innate Immune Component

#### 3.1.1. Tumor-Associated Macrophages (TAMs)—Microglia and Bone Marrow Derived Macrophages (BMDM)

Approximately 30–40% of the TME are innate immune cells called tumor- associated macrophages (TAMs) [31,32]. They are formed by two distinct macrophage populations—microglia and BMDM [33,34]. 

Microglia are resident macrophage-like cells that are developed from the yolk sac during embryogenesis [35]. They are the only immune cells in the brain at steady state and act as both sentinel immune cells and regulators of homeostasis [36]. Their survival depends on stimulation from colony-stimulation factor (CSF) 1 or interleukin (IL) 34 via its receptor, CSF1R. Although microglia are the only macrophages in a naïve brain, they only compose approximately 15% of the macrophages [37] and reside on the periphery of the TME [38]. The rest of the macrophages found in the TME are BMDM.

BMDM do not exist in the brain at steady state. They are circulating monocytes originating from hematopoietic stem cells within our bone marrow or spleen. Upon inflammation, monocytes can migrate to the sites of infection/inflammation and differentiate into macrophages [36,39,40]. Likewise, in gliomas, monocytes can infiltrate via chemotaxis [41,42], taking advantage of the breakdown of the Blood Brain Barrier (BBB) during glioma pathogenesis [43,44,45,46] and through receptor stimulation, differentiating into macrophages. Unlike microglia, BMDM are found intratumorally [38,47]. Within the TME, BMDM exist on a spectrum of polarization between M1 (pro-inflammatory/immune-stimulatory) and M2 (anti-inflammatory/ immunosuppressive). Glioma cells recruit peripheral monocytes and polarize them toward the M2 phenotype. The degree of BMDM recruitment correlates positively with glioma grade and progression [34] and negatively with prognosis [48].

#### 3.1.2. Neutrophils (PMNs)

Although neutrophils are the most abundant leukocytes in our blood stream, they are not a major component within the glioma TME. Under steady state, they are generated from haemopoietic stem cells from our bone marrow. During infection, neutrophils can exert multiple actions, such as phagocytosis, degranulation, release of neutrophil extracellular trap, and antigen presentation, in order to control the spread of foreign invaders [49]. Like BMDM, neutrophils correlate with prognosis negatively, and their role in gliomas is starting to be recognized [50]. A recent study identified a unique population of myeloperoxidase (MPO)-positive macrophages associated with long-term survival [51]. Neutrophils express abundant MPO, and one could infer that these macrophages could be engulfing neutrophils, but this may reflect a more complex process.

#### 3.1.3. Dendritic Cells 

Dendritic cells (DCs) are a diverse group of myelocytes that are well known for their ability to survey nearby environments, uptake and process antigens, and activate T cells. For this, they are known as one of the professional antigen-presenting cells (APCs), linking our innate and adaptive immune systems, and they exist within the glioma TME [52,53]. Their role in the TME is not yet elucidated, but there seems to be a positive correlation between the frequency of TILs with DCs within the TME [52].

#### 3.1.4. Natural Killer (NK) Cells

NK cells are innate cells with cytotoxic capabilities, and their presence in the glioma TME has been described [54]. Unlike cytotoxic T cells (discussed later), NK cells detect targets for killing by a mechanism called “missing self” instead of antigenic stimulation. Foreign invaders, such as bacteria or viruses, can downregulate MHC class I molecules and avoid cytotoxic T cell recognition. NK cells can detect the downregulation of MHC class I molecules and are actioned to kill. Tumor cells can downregulate MHC class I molecules as a mechanism of immune escape, but NK cells can potentially kill these tumor cells. It has been shown that chemokines secreted by glioma cells can attract NK cell infiltration to the TME, and this is associated with better prognosis [55].

#### 3.1.5. Monocyte-Derived Suppressor Cells (MDSCs)

MDSCs are a heterogenous group of cells with their main function of putting a break on our immune system. There are two main types of MDSCs—monocytic and polymorphonuclear (PMN). Both types are found in the glioma TME [56], and their numbers correlate negatively with prognosis [57].

### 3.2. Adaptive Immune Component

#### Tumor-Infiltrating lymphocytes (TILs)

TILs found in the TME consist of NK (described above), CD4+ T, CD8+ T, and B cells. CD4+ T cells, also known as helper T cells, orchestrate our adaptive immune system. They are activated by professional APCs and differentiate into T helper 1 (Th1), T helper 2 (Th2), and T helper 17 (Th17) cells depending on the stimulating cytokines. Th1 cells can polarize the environment towards cellular immunity (CD8+ T cell response). When activated, CD8+ T cells cause cellular damage to target cells. Thus, they are also known as cytotoxic T cells. They kill cells via multiple mechanisms—cytokine (IFN-g and TNF-a) secretion, FAS-ligand-receptor signaling, and perforin and granzyme release [58]. Th2 cells can polarize the environment towards B cell-mediated humoral immunity (antibody response). Th17 cells are usually associated with autoimmunity but may also play an anti-tumoral role [59]. 

Unlike the abundance of TAMs in the glioma TME, TILs are scarce and comprise only 0.25% of the cells. T cells seem to home to bone marrow rather than the TME [60]. Of these, the majority are functionally exhausted and ineffective [61,62]. Furthermore, anti-tumor lymphocytes are drastically reduced (25% of the already depleted lymphocyte population) [63]. This contrasts heavily with tumors with a favorable immunotherapy response which have a comparatively higher number of lymphocyte infiltration. Regulatory T cells are also TILs found in the glioma TME. Unlike the other lymphocytes, regulatory T cells are immunosuppressive and confer a poor prognosis in glioma patients.

## 4. Non-Tumoral, Non-Immune Components of the TME

Most of these cells are comprised of glia, neurons, and cells related to the blood brain barrier apparatus. 

### 4.1. Glial Cells

Glial cells or glia comprise up to two-thirds of normal brain tissue and include astrocytes and oligodendrocytes [64,65]. Initially described as “nerve-glue” for their believed structural role, it is now appreciated that they provide numerous and varied important roles to CNS homeostasis [65]. 

Astrocytes are the most common glial cells in normal brain tissue and are most commonly found in perivascular niches [66]. In addition to general structural support, they provide general homeostatic functions, such as maintaining water and ion balances and blood brain barrier integrity, and regulating neuronal synaptic activity and immune response [67,68]. For example, astrocyte foot processes maintain the glia limitans, which is the outer layer of the blood brain barrier [69,70]. In addition to contributing to blood-barrier integrity, these processes densely express Fas-ligand, which induces apoptosis in cells with Fas-receptors. Given that activated T cells express Fas receptors, astrocytes, therefore, limit CNS entry of T cells [69]. Other examples of CNS immune regulation by astrocytes include anti-inflammatory modulation through TGF-beta release [71]. 

Oligodendrocytes are another major glial cell. Their main role is to maintain and supply myelin sheaths to neuronal axons in the CNS [72]. 

### 4.2. Neurons

It has long been appreciated that the neuron conducts electrical signals that constitute brain function. However, data is now emerging concerning its role in the modulation of the brain microenvironment. For example, neurons express CD200, which activates myeloid, microglia, and lymphocytic cells through the CD200 [73,74].

Neurons also control the growth of CNS cells. Electrochemical modulation of oligodendroglial and neuronal precursor cell differentiation and survival are such instances of this [75,76,77,78,79]. Another emerging area of study is the role of paracrine influence from neuronal cells. An example of this is the activity-regulated release of neuroligin-3 (NLGN3), which is a synaptic cell-adhesion molecule. When NLGN3 binds to its receptor, neurexin, it connects neurons at synapses and modulates synaptic function and signaling and has been shown to manipulate normal brain parenchyma and the TME [80]. Both electrical and paracrine aspects of neuronal influence are important to note, as they can be hijacked to drive a favorable TME [13,81].

### 4.3. The Blood Brain Barrier (BBB) and Vasculature 

The BBB is the major site of blood oxygen–oxygen in the brain [82,83]. It has permeability for hydrophilic and small polar molecules while excluding larger hydrophilic molecules, providing protection to the CNS from toxins and pathogens [84,85,86].

The BBB has an inner endothelial layer closely connected by inter-cellular junctions. These tight junctions maintain strict permeability with a variety of proteins, such as claudin and occluding and adhesion molecules [87,88]. Surrounding this is the basement membrane with pericytes and astrocytic foot processes (discussed above) [89]. 

Evidence is emerging that BBB function and structure are related to an interplay between nearby cells such as pericytes and vascular smooth muscle cells, astrocytes, microglia, and neurons. Together with the BBB, these cells are termed the neurovasculature unit (NUV) [90,91,92]. An example of this is Vascular Endothelial Growth Factor (VEGF), which is strongly associated with tumor neurovasculature in glioblastoma, with virulent expression stimulating tumor angiogenesis and vascular proliferation [93]. The result of which is semi-permeable BBB with inconsistent pH, blood supply, and fluid shift, making drug delivery to the tumor very hard to predict.

### 4.4. The Extra-Cellular Matrix

The ECM consists of a combination of interstitial fluid and minerals and a variety of proteins. These proteins include collagen and elastin, which provide structural support, glycoproteins such as fibronectin, laminin, and tenascin, as well as proteoglycans and glycosaminoglycans [94]. The proportion of fibrillar to non-fibrillar components varies between tissue types. In the brain, the ECM has much higher concentrations of glycosaminoglycans, including hyaluronic acid and proteoglycans, such as heparan sulfate and chondroitin sulfate [95]. However, this composition differs in the ECM of gliomas [95]. Studies have demonstrated that the glioma ECM consists of higher concentrations of collagen compared to the normal brain and that collagen expression increases with glioma grade [96,97].

## 5. Tumoral Influence on TME

Tumoral influence on the TME is potentially the most studied component of TME interactions. Perhaps the most novel sphere of influence is that of electrical/inter glial cell communication. This is of particular importance as it highlights new therapeutic targets to disrupt the glial network. In addition, the current view of glial effect on the TME is one of variation depending on the oncogenic molecular pathways and the transcriptomic signatures of the tumor cells [98]. 

### 5.1. Tumoral Electrical Signaling

Some glioma cells produce rhythmic calcium-dependent electrical signals that propagate through the glioma network via synapses. These cells are characterized by the KCa3.1 protein and are believed to drive tumor networking through these impulses [99]. They demonstrated on single-cell RNA-sequencing that these generator cells represent a very small number of glioblastoma cells (1–5%) and that these cells were enriched with a mesenchymal transcriptomic signature [99]. These cells tend to locate away from the advancing tumor edge and are well networked with synapses between other glioma cells. However, unnetworked cells at the periphery of the TME become stationary over time and develop tumoral networks [99]. 

The implication of this process is still being elucidated, but glioma cell-initiated electrical propagation has been shown to activate oncogenic pathways (MAPK and N NF-κB) and is associated tumor growth and microglia activation [99,100,101].

### 5.2. Effect of GSC on TME

Although the primary role of GSCs is to provide a cell reservoir to repopulate the tumor cell population, it is now appreciated that they also have a direct impact on TME manipulation. As they predominantly reside in the perivascular niche, they are ideally situated to alter the vasculature component of the TME and are known to release a variety of factors to increase neovasculature and subsequent tumor sphere formation [4,19]. They also have the ability to differentiate into vascular endothelial cells themselves, further increasing angiogenesis capabilities [5,102,103].

Monocyte recruitment and polarization toward the M2 subtype are also increased by GSCs through the release of chemokines such as CCL2 and CSF-1 [104].

### 5.3. Influence of Tumor Molecular Phenotype on the Immune Component

Understanding the immune impact of tumor phenotype is crucial to developing future treatments for gliomas. Here we describe tumor-immune interactions through transcriptional and molecular phenotypes/oncogenic pathways and *IDH1/2* status. 

#### 5.3.1. Influence of Transcriptional Signatures on Immune Component 

Single-cell RNA-sequencing has shown heterogeneous expression of transcriptional subtypes and variation in gene expression, including those associated with an immune response [105,106]. Variation in immune response has also been shown to be present when using transcriptional glioma subtyping [107]. For example, the classical glioblastoma subtype was associated with DC when compared to proneural (associated with CD4+ gene expression) and mesenchymal (decreased NK but increased M1 macrophage and neutrophils expression) [107,108]. 

Furthermore, the mesenchymal subtype has been shown to have an overall increased immune presence compared to the other subtypes [109,110]. They have the most tumor-infiltrating CD3+ and CD8+ T cells, with proneural tumors having the lowest [109].

#### 5.3.2. Influence of Oncogenic Pathways on Immune Component

A major contributing factor to strong tumoral influence on the TME is the crossover of oncogenic pathways that drive anti-inflammatory characteristics. These pathways result in cytokine, chemokine, and receptor/ligand expressions that create an anti-inflammatory TME phenotype. A variety of anti-inflammatory mechanisms have been linked to common oncogenic pathways found in GB. These include mutations in PI3K, Ras-MAPK, WNT/Beta-catenin, and p53 pathways [111]. 

One of the most commonly activated oncogenic pathways in gliomas is the Ras-MAPK pathway through NF-1 alterations [112]. This pathway, in turn, leads to multiple immune-modulatory processes, including IL6 production which, in turn, leads to CCL2 expression and macrophage recruitment (see Figure 2) [37,113]. Macrophage recruitment then allows for polarization toward the anti-inflammatory M2 phenotype. Polarization can occur through prolonged exposure to IL6, as well as to the anti-inflammatory cytokine TGF-Beta. The latter is also produced by Ras-MAPK activation through p38 MAP kinase activation [114]. These factors have also been shown to inhibit DC and lymphocyte migration into the TME [115].

PI3K activation commonly occurs through aberrations in EGFR and c-met pathways and is shown to increase the expression of PD-1 receptors on T cells [116]. This subsequently increases T cell exhaustion and impairs adaptive immune response in the TME. 

These pathways converge on the transcription factor STAT3. In gliomas, STAT3 activation is associated with glioma genesis and transformation to the mesenchymal phenotype. Mechanisms of activation include EGFR and PDGFR activation and even cytokine stimulation such as IL6 (see Figure 2) [117]. In turn, STAT3 activation further promotes monocyte recruitment through CXCL1 and CXCL2 expression [118], which further adds to the pool of TAMs to be polarized toward the anti-inflammatory phenotype. 

There is an association of oncogenic pathways and higher glioma grades with CSF-1 expression [119]. It is believed that this overexpression plays a strong role in polarizing TAMs recruited by the above process toward the M2 phenotype. This, in turn, blunts the immune response in the TME, as discussed below [119,120,121,122]. 

#### 5.3.3. Influence of IDH1/2 Status on Immune Component of TME 

Immune suppression in the TME of *IDH1/2*-mutant gliomas may come from the presence of 2-HG. In mouse models, the addition of the *IDH* gene or exposing glioma cells to 2-HG led to a reduction in CD8 cytotoxic T cells and an expression of cytotoxic T cell-associated genes such as CXC ligand. Furthermore, inhibiting IDH in these models led to better T cell recruitment [123]. In addition, there is evidence that 2-HG impairs T cell function. This may be explained by the effect of 2-HG on T cell receptor signaling, which prevents T cell activation [124]. 2-HG further interferes with the activation of the adaptive immune system by reducing the expression of costimulatory molecules CD80 and CD86 and MHC class-II [125].

In a similar fashion, IDH1/2 mutations lead to the loss of NK cell-mediated cytotoxicity. 2-HG may suppress activating receptor, NKG2D ligands, and ULBP1 and ULBP3 genes (see Figure 3) [126]. The presence of 2-HG is also associated with the reprograming of TAMs toward the M2 phenotypes while increasing the release of anti-inflammatory cytokines IL-10 and TGF-beta [125]. Immune suppression of the IDH-mutant TME is also related to a reduction in immune cell recruitment, characterized by a reduction in chemotaxis factors compared to *IDH1/2* wildtype gliomas [123,127]. 

Differences in immune manipulation reflect on different immune TME between *IDH1/2* mutant and wildtype gliomas. While the most abundant immune cell is M2 polarized TAM in both cases [128,129], immune cells, in general, are much more abundant in *IDH1/2* wildtype gliomas [130]. There are also lower numbers of DC and immune suppressor cells, such as Tregs in IDH-mutant gliomas, especially in oligodendrogliomas [131].

Deconvolution analysis of bulk RNA-sequencing has shown that M0 macrophages were increased in IDH-wild; however, monocytes were more common in IDH-mutant gliomas [132]. Bunse et al. showed that there was a reduction in T cell abundance in IDH-mutant gliomas. Those that were present were enriched for CD4+ naive T cells and had a reduction in memory T cells [124]. 

#### 5.3.4. Tumoral Influence on Non-Immune TME Components

Glioma cells influence non-immune TME constituents via the vascular endothelial growth factor to drive angiogenesis and endothelial cell proliferation [133]. This results in aberrant and dysfunctional blood vessel formation in the TME [134].

Another prominent non-immune TME component influenced by glioma cells is astrocytes. Glioma cells condition astrocytes to support tumor growth. This is done through releasing factors such as the Receptor activator of nuclear factor kappa B ligand (RANKL). The activated astrocytes consequently release a variety of growth factors driving tumor growth (discussed in further detail below) [135,136]. 

Glioma cells have also been shown to directly alter the ECM of the TME. Not only does the ECM constituents change with glioma grade, but there are translational reports of glioma cells producing type I collagen [96]. 

### 5.4. Immune Influence on TME

The immune system undisputedly shapes the glioma TME. The various immune cells described in Section 3 can promote tumor growth and progression or tumor elimination. As alluded to in earlier sections, chemokines, cytokines, and growth factors are important in drawing immune cells to the glioma TME and polarizing immune cells into cells with either pro-inflammatory or anti-inflammatory properties. 

Microglia are likely the first immune cells to encounter glioma cells, given they are tissue resident macrophages. They express the chemokine receptor CX3CR1 and is attracted to the glioma periphery by CX3CL1 [34]. Due to the anti-inflammatory environment in gliomas, and the secretion of CSF-1 in the TME, microglia predominantly polarize towards M2 phenotype [120]. These microglia can release chemokines such as CCL2 [137], MIP-1, CCL3, and CCL5 [138] to attract peripheral monocytes into the TME. With the aid of the anti-inflammatory cytokines and CSF-1 secreted by both microglia and glioma cells, infiltrating monocytes can differentiate into BMDM and polarize toward a M2 phenotype [120,139]. Due to the abundance of TAMs within the TME, they can physically limit the number of TILs in contact with glioma cells [140]. Furthermore, TAMs are shown to suppress T cell response normally by secreting indoleamine pyrrole-2,3-dioxygenase (IDO), IL-10, and TGF-b [136,141,142]. Although macrophages are commonly known as one of the antigen-presenting cells with the ability to prime antigen-specific T cells, TAMs are unlikely to generate effective T cell response. There are several reasons for this. One, tissue resident macrophages are programed to clear apoptotic cells without inducing the immune system [143]. Toll-like receptors (TLRs) are shown to be reduced in TAMs [144], rendering their ability to recognize danger-associated molecular patterns (DAMPs), induce the formation of long-term memory T cells, and avoid T cell exhaustion [145,146,147]. Two, immunoproteasomes, important for generating immunogenic peptides needed to activate effect T cell response, may not be induced in TAMs due to the low expression of IFNs within the TME [148,149,150,151]. Three, TAMs are known to have low levels of MHC class II molecules and, therefore, unlikely to induce effective T cell response [152].

MDSC can also be recruited to the TME and incorporated into TAMs [153], causing immunosuppression. Although they can have suppressive effects on various immune cells, such as NK cells [154], macrophages, and DCs [155], their major targets are T cells [156]. They can render effector T cells ineffective by producing anti-inflammatory cytokines such as IL-10 [157], upregulating inhibitory molecules such as PD-L1 and CTLA-4 [158,159], causing memory T cell dysfunction [160], and inducing Tregs [61,161], which can cause T cell exhaustion and death [62]. Furthermore, MDSC can also support tumor growth and migration [162] and provide a pro-survival environment for cancer stem cells [163].

Neutrophils and DCs are also recruited to the glioma TME together with BMDM. Most studies have shown that neutrophils play a detrimental role as they are associated with tumor progression, promote GSC survival, and facilitate angiogenesis [50,155,164,165]. Recently, there is evidence that neutrophils can infiltrate early-stage tumors and limit tumor growth [166]. However, the anti-tumor effect is limited to the early establishment of glioma cells. Neutrophils entering the TME at later stages seem to be immature and equipped with the ability to suppress T cell function [166]. DCs are pivotal in the activation of T cells and mount anti-tumor immunity. Their presence in the TME positively correlates with TIL numbers [52]. There are two types of DCs found in the glioma TME—plasmacytoid DC and conventional DC. Plasmacytoid DCs produce IFN-a, a cytokine that has been shown to improve the survival of high-grade glioma patients [167]. Conventional DCs are well known for their ability to capture and process tumor antigens for priming of anti-tumor T cells [168,169,170]. Unfortunately, the downregulation of TLR and the presence of anti-inflammatory cytokines in the glioma TME negatively impact the ability of these DCs to prime effective anti-tumor T cell responses.

Tumor cytotoxicity is mainly mediated by NK cells and cytotoxic T cells. NK cells have been shown to have the ability to target glioma stem cells [171], and anti-glioma cytotoxic T cells response has been shown to be inducible and associated with better prognosis [172]. Unfortunately, most of these cytotoxic cells are dysfunctional [62,62,154], and much work has been done to reverse this.

## 6. Non-Tumoral, Non-Immune Influence on the TME

The components of this category with significant influence on the TME are glial cells (particularly astrocytes), neurons, and the BBB.

### 6.1. Influence of Glial Cells on the TME

Astrocytes alter their phenotype on exposure to glioma cells into what is termed active astrocytes [173]. This phenotype is characterized by the expression of GFAP protein. 

Active astrocytes release a variety of factors that shape the TME to be more favorable to glioblastoma. Factors released by active astrocytes include cytokines, matrix metalloproteinases, and other growth factors such as insulin-like growth factor 1 (IGF-1). They also directly affect gene regulation through gap junction communication with glioma cells [135,136,174,175,176]. The overall effect is increased glioma growth and treatment resistance [177]. Release of metalloproteinase also helps tumor invasion into normal brain tissue [176]. 

Activated astrocytes also have a strong influence on the immune component of the TME. They have been shown to secrete multiple cytokines, such as TNF-alpha, TGF-beta, IL-10, and IL-6 [135,136]. This cytokine release shown in CNS metastatic models further contributes to cell survival and treatment resistance through upregulation of STAT1, NFκB, GSTA5, BCL2L1, and TWIST1 [178]. Furthermore, in metastatic cancer cell models, astrocytes release miR-19a-containing exosomes that inhibit PTEN, which further supports cell growth [179].

The effect of oligodendrocytes on the TME is much less clear than astrocytes. However, it has been shown that the number of oligodendrocytes in *IDH1/2* wildtype glioblastoma is much higher than in mutant cases. This signals a likely tumor–oligodendrocyte communication that remains to be elucidated [180].

### 6.2. Effect of Neurons on the TME

Neurons have been shown to manipulate the TME primarily by driving cell growth [181]. Venkatarmani et al. showed that neuronal activity increased the rate of glioma cell network branching [13]. Synapses form between glioma cells and neurons, and that glioma depolarization through these synapses leads to cell growth (Figure 4) [182]. It was also shown that synaptic formation was increased on exposure to neuroligin-3 (NLGN3) when cleaved and released by ADAM10 (See Figure 4). NLGN3 has been shown to have a paracrine influence on both neurons and glioma cells. NLGN3 activates a variety of pathways shown to favor glioma growth, such as the AKT-mTOR pathway and MAPK pathway (see Section 2.2 Mutational landscape of gliomas) [183]. Other neuronal-related paracrine/autocrine singling include AMPA and glutamate, which have been shown to also drive glioma growth [184,185]. It is now appreciated that glioma cells produce microtubes that are able to establish synapses with neurons, referred to as neuroglial synapses. These synapses produce postsynaptic currents initiated through the glutamatergic AMPA receptors. The electrical stimulation from these synapses, in turn, drives glioma cell growth [186]. Adding to this, there is strong evidence that neuronal activity increases oligodendrocyte precursor cell growth and proliferation. Given oligodendrocyte precursor cells are a strong candidate for a cell of origin for gliomas, it is likely that neuronal activity can drive glioma genesis as well as glioma growth [181,187].

### 6.3. Effect of BBB and Vasculature on the TME

As discussed above, due to tumoral compromise of the blood brain barrier, there is increased permeability from aberrant blood vessel formation of the BBB in the TME [134]. This, in turn, leads to an effect on the TME. Due to the high degree of permeability of glioma-associated blood vessels, there is increased interstitial pressure, which then drives hypoxia and necrosis [188]. This hypoxic environment then drives macrophages toward the anti-inflammatory M2 phenotype through Sema/Nrp1 signaling [189]. 

The BBB/tumor-associated vascular also directly influences the TME. When activated by VEGF signaling from glioma cells, endothelial cells release growth factors, such as TGF-beta, FGF, and EGF, which support tumor cell growth. These factors, in particular, nurture GSC growth [133]. Similarly, nitric oxide released from endothelial cells supports GSC [190]. 

CTL isolated from the TME has an exhausted phenotype limiting the anti-tumor immune response [62]. In the presence of inflammatory cytokines such as interferon-gamma (IFN-γ) and tumor necrosis factor-alpha (TNFα), the BBB vasculature at the local tumor site becomes “activated”, leading to upregulation of PDL1/2 ligands that increase T cell exhaustion [191,192]. 

### 6.4. Effect of the ECM on the TME

Perhaps the ECM’s most recognized effect on the TME is facilitating tumor cell migration [193,194]. However, it has also been shown to regulate the infiltration of immune cells into the TME [195,196]. In addition to its structural/migration effect on the TME, the ECM can also directly influence the cellular components of the TME. The ECM components have been shown to influence protein and mRNA expression in cells through contact with cell receptors [197]. For example, fibronectin can induce TGF-beta expression and suppress p53-driven apoptosis in glioma cells [198,199]. Furthermore, breakdown products from the remodeling of type 1 collagen fibers act as chemoattractants to immune cells such as neutrophils [200,201]. Collagen fibers can also limit the cytotoxic effect of natural killer cells by activating the inhibitory receptor LAIR-1 [202]. Conversely, collagen can lead to the pro-inflammatory change of certain immune cells such as neutrophils through activation of the immune receptor OSCAR [203,204]. 

## 7. Future Therapeutic Directions

The glioma TME environment represents a network of complex interactions between molecular pathways of the glioma cells and other TME components. These interactions result in a highly anti-inflammatory immune phenotype. Therefore, novel treatment options can arise from either targeting TME singling and networking or reversing the immunosuppressive environment.

### 7.1. Interruption of TME Signaling

Perhaps the most straightforward approach to disrupting TME signaling is to directly target the molecular pathways driving glioma activity and TME interaction. Unfortunately, this has been unsuccessful so far. Numerous trials have attempted to block EGFR through a variety of mechanisms but have not yielded significant results [205], and blockade of the Rb-pathway, using palbociclib, was unable to demonstrate efficacy. Trials targeting the AKT-mTOR pathway, such as those using the pan-PI3K inhibitor buparlisib or the mTOR inhibitor temsirolimus, have also been negative [206,207]. This may be explained by the highly plastic nature of glioma cells and their ability to mutation switch within individual molecular pathways [22]. 

A potential avenue around this is to disrupt glioma signaling by targeting the transcription factors activated by these oncogenic pathways. A prime target is the transcription factor STAT3. Not only is it used in glioma cell growth and immune signaling, but it is also a key driver in the activation of astrocytes. The STAT3 inhibitor silibinin reduces astrocyte activation and reduces rates of brain metastases [208]. When administered to 18 patients with lung cancer brain metastases, STATs inhibition increased overall survival. Such findings therefore show promise in the glioma setting [208]. 

One potential strategy to interfere with glioma TME signaling is to disrupt the electrical signaling in the glioma network. One example would be the inhibition of KCa3.1. Given this protein is required for calcium-dependent signal propagation through the glioma network, its blockade has led to a reduction in glioma invasion and activation of microglia [99,100,209]. Another option to disrupt electrical signaling is by blocking the neuroglial glutaminergic synapses. It has been shown preclinically that glioma cell growth can be perturbed through the use of the anti-epileptic and anti-AMPA receptor perampanel. The drug is currently used in the clinical setting for the treatment of seizures and could be easily adapted to be used in glioma-focused clinical trials.

Another novel target that can interrupt TME signaling is ADM10. It has been shown that inhibition of ADAM10, which allows the release of NLGN3 into the TME, leads to reduced levels of NLGN3 and growth inhibition of xenograft animal models. Given NLGN3 activates multiple pro-glioma molecular pathways, its reduction may disrupt crucial glioma signaling potential [183]. Excitingly, ADAM10 inhibitors have been used in clinical trials in the non-glioma setting and appear well tolerated [183,210,211]. 

### 7.2. Reverse Immunosuppression of TME

Immune checkpoint inhibitors (ICIs) have been a breakthrough in cancer treatment [10,11,212,213]. To date, ICIs have been disappointing in clinical trials for gliomas [12,214]. A major barrier is the immunosuppression within the TME. Given the huge impact of polarized M2 macrophages on the immunosuppressive effect of the TME, there is considerable focus on interrupting macrophage recruitment and polarization in the TME. Perhaps the most promising approach in this instance is CSF-1 inhibition, which has been shown to reduce glioma recurrence after radiation in vivo [215]. However, clinical trials using anti-CSF1R antibodies have yet to show clear benefits [216,217]. This is likely because CSF-1 had no impact on the phenotype of TAM once polarized [119]. There is evidence that the TME can drive resistance to CSF-1R inhibitors [218]. This explains why CSF-1R inhibition impacted microglia cells in the peripheries of the TME but had little effect on BMDM within the TME [120].

Multiple clinical trials investigating DC vaccines suggest overall survival improved [219,220,221]. Conceptually, this seems to be a promising therapy, as DCs specialize in priming anti-tumor T cell responses. A non-randomized phase III trial reported that treating patients with recurrent glioblastoma with lysate loaded DCs improved survival. Patients had better survival compared with patients in other published clinical trials who were considered as “external control” [221]. Concerns have been voiced regarding the validity of the external control and that the lysate was manufactured from the primary resection sample but used to treat recurrent glioblastoma [222]. Unfortunately, all published DC vaccine trials remain either uncontrolled or externally controlled, and the clinical utility of DC vaccines is yet to be elucidated. In order for DCs to effectively prime anti-tumor T cells and generate long-lasting memory T cells, the presence of PAMPs or DAMPs (described in Section 5.4) is crucial. Future avenues in DC therapy should include co-administrations of PAMPs or DAMPs, such as TLR agonists, in the setting of randomized controlled trials.

The scarcity of T and NK cells within the glioma TME is one of the reasons why immunotherapies, such as CTLA-4 and anti-PD-1/L1, are ineffective. Introducing anti-tumor T/NK cells can potentially overcome this problem. Engineered chimeric antigen receptor (CAR) T cell have shown benefit in hematological malignancies [223]. Their effectiveness in highly heterogeneous tumors like glioblastoma is yet to be shown. Furthermore, NK CAR is currently being engineered in murine models [224,225]. Two proteins, EGFRvIII and IL-13R, had been described to be expressed on glioma cells, and CAR T cells engineered to target these two proteins were trialed in glioma patients. Although these CAR T cells can kill glioma cells in vitro and in vivo, these have not yet shown to improve survival outcomes [226,227]. Identifying specific antigenic target to engineer the right T cell receptor can be difficult due to deadly on-target, off-tumor side effects [228], but more importantly, finding a way for T cells to break through the wall of stromal cells and a large number of BMDM crowding the TME to kill glioma cells is even more challenging. Future CAR T/NK cell therapies will require CARs targeting multiple targets combined with strategies to ensure tumor infiltration and tumor contact.

Given the strong evidence of multiple oncogenic pathways driving immune suppression, there is increased interest in inhibiting these molecular processes to reduce immune suppression of the TME. Such examples include inhibitors of the PI3K and Ras-MAPK pathways [115]. In the case of *IDH1/2*-mutant gliomas, the most tempting target for reducing immune suppression is 2=HG. Inhibition of the *IDH*-mutant gene may lead to increased T cell recruitment in mouse models [123]. Moreover, the use of an AhR inhibitor may improve the immune suppressive phenotype when given with an IDH inhibitor, compared to IDH inhibitor alone [124,125]. However, these are yet to be tested in randomized clinical trials.

## 8. Conclusions

The glioma TME is complex, and the interplay between its cellular content is important in developing effective targets for tumor elimination. So far, the most effective treatments for gliomas consist of surgery, radiotherapy, and chemotherapy. Immunotherapy and targeted therapies have been disappointing. However, most of the novel therapies studied seem to target only one or two parts of the glioma TME. Combined multimodal therapy targeting glioma signaling pathways, anti-inflammatory cytokines, and immunosuppressive cells may be the future of effective glioma therapy.

## Figures and Tables

**Figure 1 cancers-15-02856-f001:**
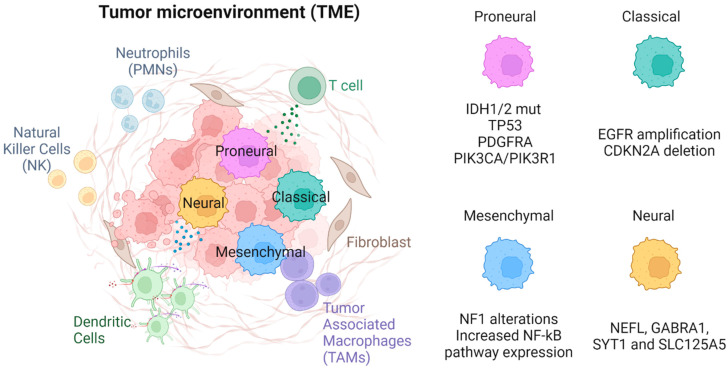
The four GBM molecular subtypes are proneural, neural, classical, and mesenchymal.

**Figure 2 cancers-15-02856-f002:**
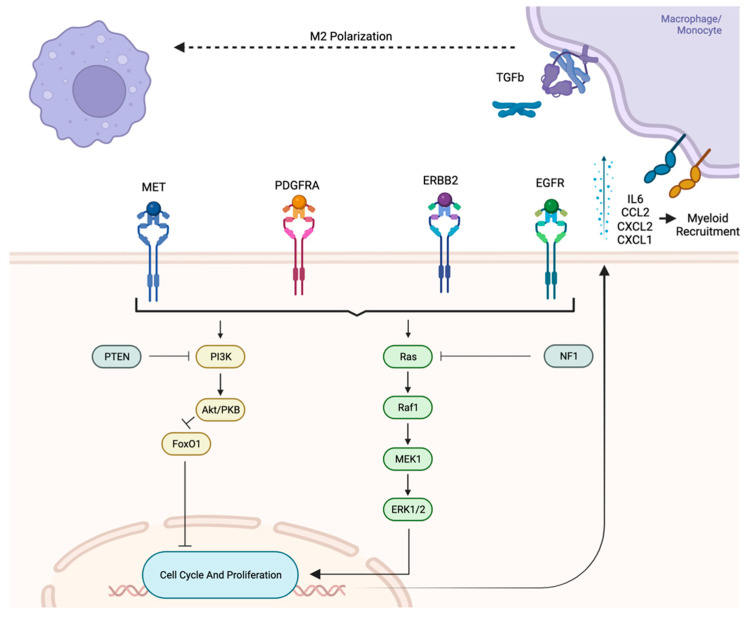
Both PI3K and Ras pathways, triggered by Receptor Tyrosine Kinases (RTKs), facilitate the recruitment of myeloid populations into tumor microenvironment, where they are polarized into pro-tumor populations.

**Figure 3 cancers-15-02856-f003:**
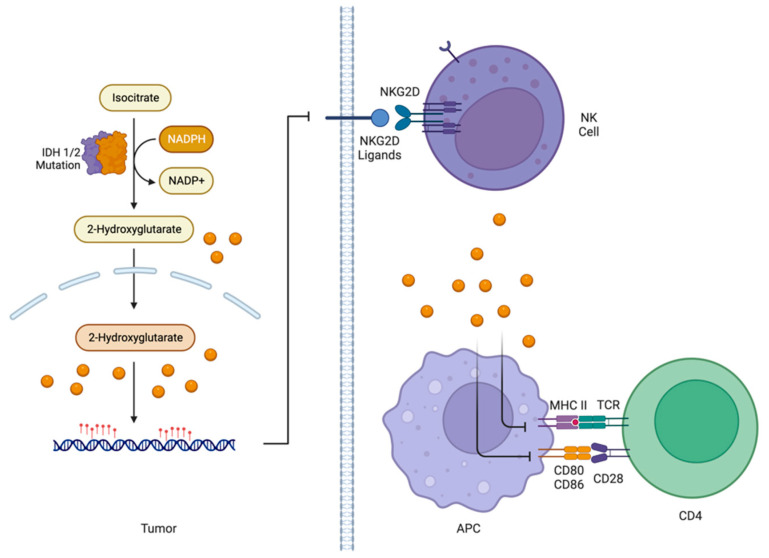
2-Hydroxygluterate produced by IDH1/2-mutant tumors may be able to evade immunity by downregulating MHC II and CD80/CD86 expression on APCs and downregulating NKG2D Ligands on the surface of tumor cells.

**Figure 4 cancers-15-02856-f004:**
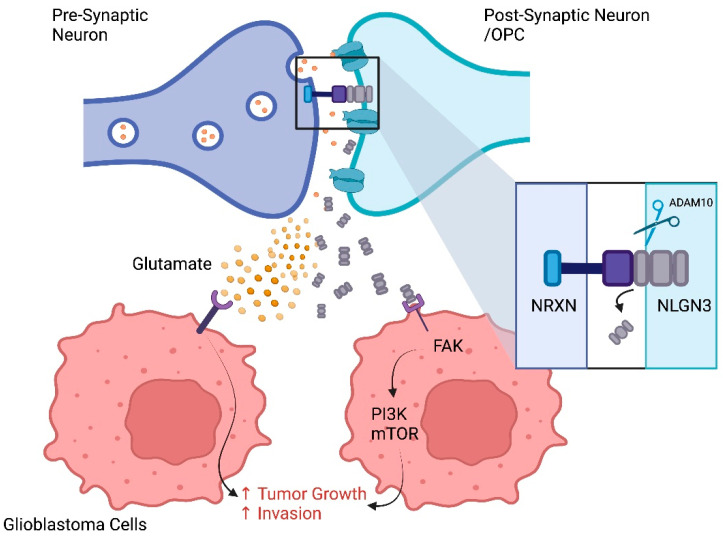
Tumor (glioblastoma) Neuronal Interactions.

## Data Availability

No new data were created or analyzed in this study. Data sharing is not applicable to this article.

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
