# Peer review of "Interdependencies of the Neuronal, Immune and Tumor Microenvironment in Gliomas"

_cancers, 2023, doi:10.3390/cancers15102856_

Round 1

Reviewer 1 Report

In the paper “Interdependencies of the neuronal, immune and tumoral microenvironment in gliomas” the authors carried out an exhaustive synthesis of the current knowledge on that issue. In the last ten years there have been significative improvements in understanding the mechanisms regulating oncogenesis and endurance of gliomas: studying the tumoral microenvironment and deepening the complex interactions between the brain cells, the immune system and the tumoral cells have brought to the awareness that there is an “interdependency” among the various actors of this drama and this has been well evidenced by the authors.  In particular, the double-cross played by the immune system has been exposed in detail. Unfortunately, the ongoing clinical trials have not produced yet sure evidences to change the standard treatment of malignant gliomas. However, so long as multiple factors contribute to the harshness of gliomas, updated multitarget treatments are warranted and a review of the available data on the complexity of the tumoral microenvironment is of paramount importance to individuate new therapeutic tools and to plan more effective interventions.

Author Response

Dear reviewers,

Thank you for your input regarding our manuscript entitled "Interdependencies of the neuronal, immune and tumor microenvironment". Please see our responses below and clean and track changes versions of the updated manuscript attached. Note all descriptions of the locations where changes have been made refer to the line numbers of the track changes manuscript version.

Reviewer 1

Comment

In the paper “Interdependencies of the neuronal, immune and tumoral microenvironment in gliomas” the authors carried out an exhaustive synthesis of the current knowledge on that issue. In the last ten years there have been significative improvements in understanding the mechanisms regulating oncogenesis and endurance of gliomas: studying the tumoral microenvironment and deepening the complex interactions between the brain cells, the immune system and the tumoral cells have brought to the awareness that there is an “interdependency” among the various actors of this drama and this has been well evidenced by the authors.  In particular, the double-cross played by the immune system has been exposed in detail. Unfortunately, the ongoing clinical trials have not produced yet sure evidences to change the standard treatment of malignant gliomas. However, so long as multiple factors contribute to the harshness of gliomas, updated multitarget treatments are warranted and a review of the available data on the complexity of the tumoral microenvironment is of paramount importance to individuate new therapeutic tools and to plan more effective interventions.

Response

We are very glad you agree with our findings of this literature review regarding tumor interdependence. We hope this review will help articulate such a complex interdependence by overviewing the various components and help prompt further interest in multi-targeted therapies to tackle this issue.

Reviewer 2 Report

The ms by Yuile et al. presents e comprehensive review of interdependencies of cellular gliomas' microenvironment. It summarizes the existing knowledge and ideas on those intercellular relations as well as presents some therapeutic perspectives.

I would have 2 recommendations for improvement:

a) as for such a review, it should include more tables/shemes/pictures (at least 2 more); the described interdependencies should be presented in quite detailed graphic form, to make the review more valuable for the readers;

b) tumor microenvironment are not only cells, but also and importantly, extracellular matrix (i.a. proteins, polysaccharides) that plays a vital role in tumor formation/progression; that aspect should also be included in the text and visualized in the ms.

Author Response

Dear reviewers,

Thank you for your input regarding our manuscript entitled "Interdependencies of the neuronal, immune and tumor microenvironment". Please see our responses below and clean and track changes versions of the updated manuscript attached. Note all descriptions of the locations where changes have been made refer to the line numbers of the track changes manuscript version.

Reviewer 2

Thank you for the excellent suggestions regarding our review, which has prompted us to make the below changes.

Comment 1

as for such a review, it should include more tables/shemes/pictures (at least 2 more); the described

interdependencies should be presented in quite detailed graphic form, to make the review more

valuable for the readers

Response 1

We agree that more detailed graphics concerning the complex subject of interdependence will increase the value of our review and help improve the readers’ understanding. This is especially so given the nature of this general overview prevents in-depth analysis of every aspect of glioma TME. Graphics can help with this. As such we have added two further illustrations, one after line 412 and one after line 445.

Comment 2

Tumour microenvironment are not only cells, but also and importantly, extracellular matrix (i.a.

proteins, polysaccharides) that plays a vital role in tumour formation/progression; that aspect should

also be included in the text and visualized in the ms.

Response 2

We also acknowledge that the omission of non-cellular components does detract from the theme of TME interdependence. As per your advice, we have attempted to rectify this by adding an overview of the ECM contents and their interplay with the remaining TME. Regrettably, we are unable to discuss the glioma ECM in detail but have given examples of its role in the overall picture to emphasize the complex interdependence of the TME.

We have added two extra sections to discuss the ECM. The first, entitled “4.4 The Extra-Cellular Matrix”, describes the extracellular TME contents and is seen on lines 329-337:

“The ECM consists of a combination of interstitial fluid and minerals and a variety of proteins. These proteins include collagen and elastin which provide structural support, glycoproteins such as fibronectin, laminin, and tenascin, as well as proteoglycans and glycosaminoglycans[95]. The proportion of fibrillar to non-fibrillar components vary between tissue types. In the brain, the ECM has much higher concentrations of glycosaminoglycans, like hyaluronic acid and proteoglycans such as heparan sulfate and chondroitin sulfate[96].  However, this composition differs in the ECM of gliomas[96]. Studies have demonstrated that the glioma ECM consists of higher concentrations of collagen compared to the normal brain and that collagen expression increased with glioma grade[97,98]”

The second, titled “6.4 Effect of the ECM on the TME”, describes the interactions of these extracellular contents with the rest of the TME and is seen on lines 588-599:

“Perhaps the ECM’s most recognized effect on the TME is facilitating tumour cell migration[195,196]. However, it has also been shown to regulate infiltration of immune cells into the TME as well[197,198]. In addition to its structural/migration effect on the TME, the ECM can also directly influence the cellular components of the TME. The ECM components have been shown to influence protein and mRNA expression in cells through contact with cell receptors[199]. For example, fibronectin can induce TGF-beta expression and suppresses p53 driven apoptosis in glioma cells[200,201]. Furthermore, breakdown products from remodelling of type 1 collagen fibers act chemoattractant to immune cells, such as neutrophils[202,203]. Collagen fibers can also limit the cytotoxic effect of natural killer cells through activating the inhibitory receptor LAIR-1[204]. Conversely collagen can lead to proinflammatory change of certain immune cells such as neutrophils through activation of the immune receptor OSCAR[205,206].  

We have also given examples of the influence of glioma cells on the ECM- lines 460-462:

“Glioma cells have also been shown to directly alter the ECM of the TME. Not only does the ECM constituents change with glioma grade, but there are translational reports of glioma cells producing type I collagen.”  

Reviewer 3 Report

The manuscript by Yuile et al., attempted to summarize the complex interplay of the TME key players in gliomas. The introduction of the TME is adequately covered.

Nevertheless, when discussing gliomas and TME, the usual context is usually in-depth. Without specifying the models and focus area,  mutations, and which aspects of TME, the scope can be too broad. This is one aspect that I would strongly suggest that authors focus on to improve the entire content and, ultimately, the aim of this manuscript. Coupled with the simple statement that mentioned mutations and the introduction to GBM subtypes, the current review seems to try to cover the various TME, but too superficial to some extent. The much-anticipated mutations and differences in genetic signature in different gliomas are vaguely discussed or, perhaps to some extent, were left out.

Considering gliomas - would authors specify the review analyzes the TME and its interaction in the pediatric or adult setting? Certainly, the interplay of the TME and its components and mutations can differ greatly. This specification or differentiation in this review work would better emphasize its focus. In the current review, the depth of the TME discussion regarding the difference key players and mutations seems underanalyzed. 

The later part of the review seems to focus on IDH mutation, and if so, the entire aim and focus of the manuscript can be directed towards IDH mutation vs. wild type, for instance. And associating them with the key mutations in gliomas. Unfortunately, this part is very much anticipated and was not well discussed and analyzed. 

Also, when discussing neuronal and gliomas. Significant work has been published in the last decade, and the current write-up is under-analyzed. What about other neuronal systems in gliomas-TME? Some references that may be useful, and the list is not limited to this:

Gillespie, S. and Monje, M., 2018. An active role for neurons in glioma progression: making sense of Scherer’s structures. Neurooncology, 20(10), pp.1292-1299. 

Broekman, M.L., Maas, S.L., Abels, E.R., Mempel, T.R., Krichevsky, A.M. and Breakefield, X.O., 2018. Multidimensional communication in the microenvirons of glioblastoma. Nature Reviews Neurology, 14(8), pp.482-495.

 Lim, S., Kim, D., Ju, S., Shin, S., Cho, I.J., Park, S.H., Grailhe, R., Lee, C. and Kim, Y.K., 2018. Glioblastoma-secreted soluble CD44 activates tau pathology in the brain. Experimental & molecular medicine, 50(4), pp.1-11. 

Venkataramani, V., Tanev, D.I., Strahle, C., Studier-Fischer, A., Fankhauser, L., Kessler, T., Körber, C., Kardorff, M., Ratliff, M., Xie, R. and Horstmann, H., 2019. Glutamatergic synaptic input to glioma cells drives brain tumour progression. Nature, 573(7775), pp .532-538. 

Venkatesh, H.S., Tam, L.T., Woo, P.J., Lennon, J., Nagaraja, S., Gillespie, S.M., Ni, J., Duveau, D.Y., Morris, P.J., Zhao, J.J. and Thomas, C.J., 2017. Targeting neuronal activity-regulated neuroligin-3 dependency in high-grade glioma. Nature, 549(7673), pp.53 3-537.

Gibson EM, Purger D, Mount CW, Goldstein AK, Lin GL, Wood LS, et al. Neuronal activity promotes oligodendrogenesis and adaptive myelination in the mammalian brain. Science. 2014;344(6183):1252304.

Glasgow SM, Zhu W, Stolt CC, Huang TW, Chen F, LoTurco JJ, et al. Mutual antagonism between Sox10 and NFIA regulates diversification of glial lineages and glioma subtypes. Nat Neurosci. 2014;17(10):1322-9.

Monje M, Mitra SS, Freret ME, Raveh TB, Kim J, Masek M, et al. Hedgehog-responsive candidate cell of origin for diffuse intrinsic pontine glioma. Proc Natl Acad Sci U S A. 2011;108(11):4453-8.

Liu C, Sage JC, Miller MR, Verhaak RG, Hippenmeyer S, Vogel H, et al. Mosaic analysis with double markers reveals tumor cell of origin in glioma. Cell. 2011;146(2):209-21.

Südhof TC. Neuroligins and neurexins link synaptic function to cognitive disease. Nature. 2008;455(7215):903-11.

Based on the analysis from the review, can authors suggest which parts of the TME and its interactions with the mutations as promising future avenues in gliomas and how? Also, can authors correlate and provide some analysis of those studies using immune cells such as dendritic cells and NK cells in gliomas and their clinical efficacy and the mutations? Following this, how would authors further suggest their improvement or future perspectives? 

To note, lines 436-437 - the citation is incorrect. Perhaps it is suggested that authors recheck the entire manuscript since the amount of references is vast. 

Author Response

Dear reviewers,

Thank you for your input regarding our manuscript entitled "Interdependencies of the neuronal, immune and tumor microenvironment". Please see our responses below and clean and track changes versions of the updated manuscript attached. Note all descriptions of the locations where changes have been made refer to the line numbers of the track changes manuscript version.

Reviewer 3

We greatly appreciate your feedback regarding our manuscript which has highlighted issues requiring further clarification. We have categorised these issues into the following sections with our changes to made below.

Comment 1

Nevertheless, when discussing gliomas and TME, the usual context is usually in-depth. Without specifying the models and focus area, mutations, and which aspects of TME, the scope can be too broad. This is one aspect that I would strongly suggest that authors focus on to improve the entire content and, ultimately, the aim of this manuscript. Coupled with the simple statement that mentioned mutations and the introduction to GBM subtypes, the current review seems to try to cover the various TME, but too superficial to some extent. The much-anticipated mutations and differences in genetic signature in different gliomas are vaguely discussed or, perhaps to some extent, were left out.

Response 1

This is an excellent point. We agree more focus on the molecular aspects of glioma will improve our manuscript. As you have pointed out the overall theme of our review is interdependence of the entire glioma TME, which differs from the conventional approach to glioma review articles on an in-depth analysis of one particular section. However, his should not occur at the expense of key research areas such as the molecular aspects of glioma. We have therefore extensively increased our discussion of mutations, genetic signatures and glioma subtypes, especially in section 2.3 Tumour cell subtypes. Examples of our additions are listed below:

Lines 121-147:

Proneural subtype gliomas are so named as they have over expression in multiple proneural development genes such as the SOX genes. They also involve genes associated with development and cell cycle/proliferation[25,26].

Glioma cells meeting the proneural signature mostly commonly had IDH1/2 mutations and as well as focal amplification of RTK receptor gene PDGFRA[22,23]. This PDGFRA amplification is seen almost exclusively in this subtype. Where PDGFRA genes are unaltered, proneural gliomas almost always have increased activity of the genes PIK3CA orPIK3R1[6]. Unlike classical subtype gliomas, proneural gliomas are also commonly associated with TP53 mutations[22].

Proneural glioma cells have an increased expression of oligodendrocyte development genes such as the afore mention PDGFRA, as well as other markers such as NKX2 and OLIG2[27]. Adding evidence to the oligodendrocyte phenotype is evidence that a protein encoded by a SOX gene, SOX10, can induce oligodendrocyte differentiation, by antagonizing the NFIA protein. Conversely, NFIA can also inhibit SOX10, leading astrocyte differentiation. Given SOX genes are over expressed in the proneural subtype, it is understandable this subtype has an oligodendrocyte-like phenotype[28].

Interestingly, OLIG2 suppresses p21 which is an apoptotic regulator of the p53 pathway[29]. The impact of OLIG2 is an example of how expression of these oligodendrocyte associated genes, are themselves tumorigenic.”

Lines 152-159:

“Glioma cells meeting the “classical” expression pattern were noted to have chromosome 7 amplification and chromosome 10 loss at a high frequency (almost 100% in classical glioblastomas). In addition, most cells had EGFR amplification and homozygous deletion of CDKN2A but were lacking TP53 and IDH mutations and PDGFRA andNF1 alterations when compared to other subtypes [6]. Other mutations commonly seen in classical type gliomas include those of NOTCH pathway such as NOTCH3, JAG1 and LFNG and in the Sonic Hedgehog pathway such as SMO, GAS1 and GLI2. This subtype is also associated with increased expression of neuronal stem cell markers[23].”

Lines 165-169:

“This subtype most notably has NF1 alterations (predominantly hemizygous deletions at 17q11.2) resulting in lower NF1 expression. There is also increased expression of genes of the tumor necrosis super family and NF-kB pathway such as TRADD, RELB and TNFRSF1A [6,24]. The increased expression of these genes may explain in part why this subtype is the most inflammatory with highest immune cell burden (discussed below).”

Comment 2

Considering gliomas - would authors specify the review analyzes the TME and its interaction in the pediatric or adult setting? Certainly, the interplay of the TME and its components and mutations can differ greatly. This specification or differentiation in this review work would better emphasize its focus.

Response 2

The review is focussed only on adult gliomas. We have attempted to make this clearer in the manuscript by adding the following to the abstract- lines 30-31:

“To this end we describe the glioma TMETME in adult gliomas through interactions between its various components and through various glioma molecular phenotypes.”

As well as to the Introduction- lines 62-64:

“Here we describe the contents of the adult glioma TME in tumoral, immune and non-tumoral non-immune components and explore its interactions between these groups.”

Comment 3

In the current review, the depth of the TME discussion regarding the difference key players and mutations seems under analyzed. 

Response 3

We agree a further in-depth analysis of the TME would greatly improve the impact of our manuscript. We have attempted to bring further analysis toward the TME with particular focus on mutations by adding an extra section in our manuscript- 2.2 Mutational landscape of glioma cells.

Lines 83-110:

In general, the most common aberrant molecular pathways are the receptor tyrosine kinase (RTK) pathways (which are further divided into the MAPK-pathway and the AKT/mTOR-pathway) the RB-pathway and the p53-pathway. Alterations in these pathways tend to be mutually exclusive and glioma cells tend to harbor an alteration in each of these pathways[20–22]. This suggests a highly interactive  network of molecular alterations.

Reflecting this interplay, it has been recognized that there are recurring patterns in the molecular and mutational landscape. For example, abnormalities of copy number variants occur at a much higher frequency than specific mutations, with deletions having a higher prevalence than amplifications[21]. The most common deletions involve the CDKN2A gene of the RB-pathway, PTEN of the AKT/mTOR-pathway and NF1 of the MAPK-pathway. While the most common amplifications include the RTK receptors EGFR and PDGFRA, MDM2 and MDM4 of the p53-pathway, PIK3CA of the AKT/mTOR-pathway and CDK4/6 of the RB pathway. The most frequently observed hotspot mutation is the TP53 gene encoding for p53. Other commonly mutated genes include PTEN, NF1 and EGFR (EGFR mutations usually occur with exon 1-8 aberrancy which is referred to as EGFRvIII)[20].

Further complicating this landscape, the glioma mutational phenotype is heterogenous and highly plastic. For example, patterns of gene mutations tend to only be seen on recurrence, such NF1 and TP53 co-mutations and Rb1 and PTEN co-deletions and LTBP4 gene abnormalities[23]. This clonal change bares consideration as they in turn can modify the tumour microenvironment, for example LTBP4 mutations have been shown to upregulate TGF-beta, which has significant anti-inflammatory properties (discussed below). It is also now recognized that mutational switching occurs in key mutational pathways such as RTK and p53 pathways, where one pathway mutation is replaced by another on recurrence[22–24].”

Comment 4

The later part of the review seems to focus on IDH mutation, and if so, the entire aim and focus of the manuscript can be directed towards IDH mutation vs. wild type, for instance. And associating them with the key mutations in gliomas. Unfortunately, this part is very much anticipated and was not well discussed and analyzed. 

Response 4

We agree that reviewing the glioma TME in the context of IDH mutant vs wildtype status would add focus to our analysis. Given we have been requested to expand on the TME interdependence by other reviewers, we are unable to fully focus the paper on IDH mutant vs IDH wildtype. This being said we have tried to expand on the differences between IDH mutant and IDH wildtype gliomas to compensate for the under analysis in this area. This occurred mainly in the discussion between differences between the proneural subtype and other subtypes, given IDH mutations are most strongly associated with the proneural subtype- Lines 121-147:

Proneural subtype gliomas are so named as they have over expression in multiple proneural development genes such as the SOX genes. They also involve genes associated with development and cell cycle/proliferation[25,26].

Glioma cells meeting the proneural signature most commonly had IDH1/2 mutations and as well as focal amplification of RTK receptor gene PDGFRA[22,23]. This PDGFRA amplification is seen almost exclusively in this subtype. Where PDGFRA genes are unaltered, proneural gliomas almost always have increased activity of the genes PIK3CA or PIK3R1[6]. Unlike classical subtype gliomas, proneural gliomas are also commonly associated with TP53 mutations[22].

Proneural glioma cells have an increased expression of oligodendrocyte development genes such as the afore mention PDGFRA, as well as other markers such as NKX2 and OLIG2[27]. Adding evidence to the oligodendrocyte phenotype is the expression of SOX10, encoded by a SOX gene, which can induce oligodendrocyte differentiation, by antagonizing the NFIA protein. Conversely, NFIA can also inhibit SOX10, leading astrocyte differentiation. Given SOX genes are over expressed in the proneural subtype, it is understandable this subtype has an oligodendrocyte-like phenotype[28].

Interestingly, OLIG2 suppresses p21 which is an apoptotic regulator of the p53 pathway[29]. The impact of OLIG2 is an example of how expression of these oligodendrocyte associated genes, are themselves tumorigenic.

We have also drawn attention to this in our analysis- lines 145-147:

“Given the strong association between IDH1/2 mutations and the proneural subtype, molecular differences between the proneural subtype compared to other subtypes, also serves as a description as the molecular phenotype associated with IDH1/2 mutations.”

Comment 5

Also, when discussing neuronal and gliomas. Significant work has been published in the last decade, and the current write-up is under-analyzed. What about other neuronal systems in gliomas-TME? Some references that may be useful, and the list is not limited to this:

Response 5

Thank you very much for this suggested reading. It is very much pertinent to our review and we have referenced this literature throughout our manuscript.

Examples include-

Lines 132-136:

“Adding evidence to the oligodendrocyte phenotype is the expression of SOX10, encoded by a SOX gene, which can induce oligodendrocyte differentiation, by antagonizing the NFIA protein. Conversely, NFIA can also inhibit SOX10, leading astrocyte differentiation. Given SOX genes are over expressed in the proneural subtype, it is understandable this subtype has an oligodendrocyte-like phenotype [28].”

Lines 303-206:

“An example of this is activity-regulated release of neuroligin-3 (NLGN3) which is a synaptic cell-adhesion molecule. When NLGN3 binds to its receptor, neurexin, it connects neurons at synapses and modulates synaptic function and signalling and has been shown to manipulate normal brain parenchyma and the TME[81].”

Lines 556-563:

“It is now appreciated that glioma cells produce microtubes that are able to establish synapses with neurons, referred to as neuroglial synapses. These synapses produce postsynaptic currents, initiated through the glutamatergic AMPA receptors. The electrical stimulation from these synapses in turn drive glioma cell  growth[188] . Adding to this, there is strong evidence that neuronal activity increases oligodendrocyte precursor cell growth and proliferation. Given oligodendrocyte precursor cells are a strong candidate for a cell of origin for gliomas, it is likely that neuronal activity can drive glioma genesis as well as glioma growth[183,189].”

Comment 6

Based on the analysis from the review, can authors suggest which parts of the TME and its interactions with the mutations as promising future avenues in gliomas and how? Also, can authors correlate and provide some analysis of those studies using immune cells such as dendritic cells and NK cells in gliomas and their clinical efficacy and the mutations? Following this, how would authors further suggest their improvement or future perspectives? 

Response 6

We have expanded on our Future Therapeutic Directions section to highlight potential therapeutic targets between glioma mutations and the TME- lines 608-624:

“Perhaps the most straight forward approach to disrupt TME signalling is to directly target the molecular pathways driving glioma activity and TME interaction. Unfortunately, this has been unsuccessful so far. Numerous trials have attempted to block EGFR through a variety of mechanisms but have not yielded significant results[207] and blockade of the Rb-pathway, using palbociclib, was unable to demonstrate efficacy. Trials targeting the AKT-mTOR pathway, such as those using the pan-PI3K inhibitor buparlisib or the mTOR inhibitor temsirolimus, have also been negative[208,209]. This may be explained by the highly plastic nature of glioma cells and their ability to mutation switch within individual molecular pathways[22].

A potential avenue around this is to disrupt glioma signalling by targeting the transcription factors activated by these oncogenic pathways. A prime target is the transcription factor STAT3. Not only is it used in glioma cell growth and immune signalling but also a key driver in the activation of astrocytes. The STAT3 inhibitor silibinin reduces astrocyte activation and reduced rates of brain metastases [210]. When administered to 18 patients with lung cancer brain metastases, STATs inhibition increased overall survival. Such findings therefore show promise in the glioma setting [210].”

We have also added a discussion regarding studies using immune cells for treatment of gliomas and suggestions for potential future directions- lines 656-695:

“Conceptually, this seems to be a promising therapy, as DCs specialize in priming anti-tumor T cell responses.  A non-randomized phase III trial reported improved survival of patients with recurrent glioblastoma who were treated with lysate-loaded DCs. Patients had better survival as compared with patients in other published clinical trials who were considered as “external control” [223]. Concerns have been voiced regarding the validity of the external control and that the lysate was manufactured from the primary resection sample but used to treat recurrent glioblastoma[224]. Unfortunately, all published DC vaccine trials remain either uncontrolled or externally controlled and the clinical utility of DC vaccines is yet to be elucidated. In order for DCs to effectively prime anti-tumor T cells and generate long-lasting memory T cells, presence of PAMPs or DAMPs (described in section 5.4) is crucial. Future avenues in DC therapy should include co-administrations of PAMPs or DAMPs, such as TLR agonists in the setting of randomized controlled trials.

The scarcity of T and NK cells within the glioma TME is one of the reasons why immunotherapies, such as CTLA-4 and anti-PD-1/L1, are ineffective. Introducing anti-tumor T/NK cells can potentially overcome this problem. Engineered chimeric antigen receptor (CAR) T cell had shown benefit in haematological malignancies [225]. Their effectiveness in highly heterogeneous tumors like glioblastoma is yet to be shown. Further more, NK CAR is currently being engineered in murine models[226,227]. Two proteins, EGFRvIII and IL-13R, had been described to be expressed on glioma cells and CAR T cells engineered to target these two proteins were trialled in glioma patients. Although these CAR T cells can kill glioma cells in vitro and in vivo, these have not yet shown to improve survival outcomes[228,229] . Identifying specific antigenic target to engineer the right T cell receptor can be difficult due to deadly on target, off tumor side effects[230],but more importantly, and finding a way for T cells to break through the wall of stromal cells and large number of BMDM crowding the TME to kill glioma cells is even more challenging. Future CAR T/NK cell therapies will require CARs targeting multiple targets combined with strategies to ensure tumor infiltration and tumor contact.”

Comment 7

To note, lines 436-437 - the citation is incorrect. Perhaps it is suggested that authors recheck the entire manuscript since the amount of references is vast. 

Response 7

Thank you for highlighting the error in referencing on lines 436-467. We have amended it to the correct reference Venkataramani et al.

Round 2

Reviewer 3 Report

The revision of this manuscript is adequate and it is ready for acceptance. 

Author Response

Thank you very mcuh